# Maximum-Entropy Adversarial Data Augmentation for Improved Generalization and Robustness

**Long Zhao**[1]    **Ting Liu**[2]    **Xi Peng**[3]    **Dimitris Metaxas**[1]

[1]Rutgers University    [2]Google Research    [3]University of Delaware

{lz311,dnm}@cs.rutgers.edu    liuti@google.com    xipeng@udel.edu

## Abstract

Adversarial data augmentation has shown promise for training robust deep neural networks against unforeseen data shifts or corruptions. However, it is difficult to define heuristics to generate effective fictitious target distributions containing "hard" adversarial perturbations that are largely different from the source distribution. In this paper, we propose a novel and effective regularization term for adversarial data augmentation. We theoretically derive it from the information bottleneck principle, which results in a maximum-entropy formulation. Intuitively, this regularization term encourages perturbing the underlying source distribution to enlarge predictive uncertainty of the current model, so that the generated "hard" adversarial perturbations can improve the model robustness during training. Experimental results on three standard benchmarks demonstrate that our method consistently outperforms the existing state of the art by a statistically significant margin. Our code is available at `https://github.com/garyzhao/ME-ADA`.

## 1   Introduction

Deep neural networks can achieve good performance on the condition that the training and testing data are drawn from the same distribution. However, this condition might not hold true in practice. Data shifts caused by mismatches between training and testing domain [6, 41, 51, 65, 76], small corruptions to data distributions [24, 75], or adversarial attacks [23, 35] are often inevitable in real-world applications, and lead to significant performance degradation of deep learning models. Recently, adversarial data augmentation [22, 51, 65] emerges as a strong baseline where fictitious target distributions are generated by an adversarial loss to resemble unforseen data shifts, and used to improve model robustness through training. The adversarial loss is leveraged to produce perturbations that fool the current model. However, as shown in [48], this heuristic loss function is insufficient to synthesize large data shifts, i.e., "hard" adversarial perturbations from the source domain, which makes the model still vulnerable to severely shifted or corrupted testing data.

To mitigate this issue, we propose a regularization technique for adversarial data augmentation from an information theory perspective using the *Information Bottleneck* (IB) [58] principle. The IB principle encourages the model to learn an optimal representation by diminishing the irrelevant parts of the input variable that do not contribute to the prediction. Recently, there has been a surge of interest in combining the IB method with training of deep neural networks [2, 5, 17, 30, 46, 60], while its effectiveness for adversarial data augmentation still remains unclear.

In the IB context, a neural network does not generalize well on out-of-domain data often when the information of the input cannot be well-compressed by the model, i.e., the mutual information of the input and its associated latent representation is high [50, 59]. Motivated by this conceptual observation, we aim to regularize adversarial data augmentation through maximizing the IB function. Specifically, we produce "hard" fictitious target domains that are largely shifted from the source domain by enlarging the mutual information of the input and latent distribution within the current

model. However, mutual information is shown to be intractable in the literature [7, 47, 53], and therefore directly optimizing this objective is challenging.

In this paper, we develop an efficient *maximum-entropy* regularizer to achieve the same goal by making the following contributions: (i) to the best of our knowledge, we are the first work to investigate adversarial data argumentation from an information theory perspective, and address the problem of generating "hard" adversarial perturbations from the IB principle which has not been studied yet; (ii) we theoretically show that the IB principle can be bounded by a maximum-entropy regularization term in the maximization phase of adversarial data argumentation, which results in a notable improvement over [65]; (iii) we also show that our formulation holds in an approximate sense under certain non-deterministic conditions (e.g., when the neural network is stochastic or contains Dropout [55] layers). Note that our maximum-entropy regularizer can be implemented by one line of code with minor computational cost, while it consistently and statistically significantly improves the existing state of the art on three standard benchmarks.

## 2 Background and Related Work

**Information Bottleneck Principle.** We begin by summarizing the concept of information bottleneck and, along the way, introduce the notations. The *Information Bottleneck* (IB) [58] is a principled way to seek a latent representation $Z$ that an input variable $X$ contains about an output $Y$. Let $I(X; Z)$ be the *mutual information* of $X$ and $Z$, i.e., $I(X; Z) = \mathbb{D}_{\text{KL}}(p_{XZ} \| p_X p_Z)$, where $\mathbb{D}_{\text{KL}}$ denotes the *KL-divergence* [34]. Intuitively, $I(X; Z)$ measures the uncertainty in $X$ given $Z$. The representation $Z$ can be quantified by two terms: $I(X; Z)$ which reflects how much $Z$ compresses $X$, and $I(Y; Z)$ which reflects how well $Z$ predicts $Y$. In practice, this IB principle is explored by minimizing the following *IB Lagrangian*:

$$\text{minimize} \left\{ I(X; Z) - \lambda I(Y; Z) \right\}, \tag{1}$$

where $\lambda$ is a positive parameter that controls the trade-off between compression and prediction. By controlling the amount of compression within the representation $Z$ via the compression term $I(X; Z)$, we can tune desired characteristics of trained models such as robustness to adversarial samples [4], generalization error [5, 11, 30, 50, 59, 64], and detection of out-of-distribution data [3].

**Domain Generalization.** Domain adaptation [21, 61] transfers models in source domains to related target domains with different distributions during the training procedure. On the other hand, domain generalization [6, 9, 39, 40, 41, 43] aims to learn features that perform well when transferred to *unseen* domains during evaluation. This paper further studies a more challenging setting named single domain generalization [48], where networks are learned using *one single* source domain compared with conventional domain generalization that requires *multiple* training source domains. Recently, adversarial data augmentation [65] is proven to be a promising solution which synthesizes virtual target domains during training so that the generalization and robustness of the learned networks to unseen domains can be improved. Our approach improves it by proposing an efficient regularizer.

**Adversarial Data Augmentation.** We are interested in the problems of training deep neural networks in a single source domain $P_0$ and deploying it to unforeseen domains $P$ following different underlying distributions. Let $X \in \mathcal{X}$ be random data points with associated labels $Y \in \mathcal{Y}$ ($|\mathcal{Y}|$ is finite) drawn from the source distribution $P_0$. We consider the following worst-case problem around $P_0$:

$$\underset{\theta \in \Theta}{\text{minimize}} \left\{ \sup_P \left\{ \mathbb{E}[\mathcal{L}(\theta; X, Y)] : D_\theta(P, P_0) \leq \rho \right\} \right\}, \tag{2}$$

where $\theta \in \Theta$ is the network parameters, $\mathcal{L} : \Theta \times (\mathcal{X} \times \mathcal{Y}) \to \mathbb{R}$ is the loss function, and $D_\theta$ measures the distance between two distributions $P$ and $P_0$. We denote $\theta = \{\theta_f, \mathbf{w}\}$, where $\mathbf{w}$ represents the parameters of the final prediction layer and $\theta_f$ represents the parameters of the rest of the network. Letting $f(\theta_f; x)$ be the latent representation of input $x$, we feed it into a $|\mathcal{Y}|$-way classifier such that using the *softmax activation*, the probability $p_{(i)}$ of the $i$-th class is:

$$p_{(i)}(\theta; x) = \frac{\exp\left(\mathbf{w}_i^\top f(\theta_f; x)\right)}{\sum_{j=1}^{|\mathcal{Y}|} \exp\left(\mathbf{w}_j^\top f(\theta_f; x)\right)}, \tag{3}$$

where $\mathbf{w}_i$ is the parameters for the $i$-th class. In the classification setting, we minimize the *cross-entropy loss* over each sample $(x, y)$ in the training domain: $\mathcal{L}_{\text{CE}}(\theta; x, y) := -\log p_{(y)}(\theta; x)$. Moreover, in order to preserve the semantics of the input samples, the metric $D_\theta$ is defined in the latent

**Algorithm 1** Max-Entropy Adversarial Data Augmentation (ME-ADA)
---
**Input:** Source dataset $\mathcal{D}_0 = \{X_i, Y_i\}_{1 \leq i \leq N}$ and initialized network weights $\theta_0$
**Output:** Learned network weights $\theta$
 1: Initialize $\theta \leftarrow \theta_0$, $\mathcal{D} \leftarrow \mathcal{D}_0$
 2: **for** $k = 1, \ldots, K$ **do**                             ▷ Run the minimax procedure $K$ times
 3:      **for** $t = 1, \ldots, T_{\mathsf{MIN}}$ **do**                 ▷ Run the minimization phase $T_{\mathsf{MIN}}$ times
 4:          Sample $(X_t, Y_t)$ uniformly from dataset $\mathcal{D}$
 5:          $\theta \leftarrow \theta - \alpha \nabla_\theta \mathcal{L}_{\mathsf{IB}}(\theta; X_t, Y_t)$
 6:      **for all** $(X_i, Y_i) \in \mathcal{D}$ **do**
 7:          $X_i^k \leftarrow X_i$
 8:          **for** $t = 1, \ldots, T_{\mathsf{MAX}}$ **do**             ▷ Run the maximization phase $T_{\mathsf{MAX}}$ times
 9:             $X_i^k \leftarrow X_i^k + \eta \nabla_{X_i^k} \left\{ \mathcal{L}_{\mathsf{CE}}(\theta; X_i^k, Y_i) + \beta h(\theta; X_i^k) - \gamma c_\theta((X_i^k, Y_i), (X_i, Y_i)) \right\}$
10:          Append $(X_i^k, Y_i^k)$ to dataset $\mathcal{D}$
11: **while** *not reach maximum steps* **do**
12:      Sample $(X_i, Y_i)$ uniformly from dataset $\mathcal{D}$
13:      $\theta \leftarrow \theta - \alpha \nabla_\theta \mathcal{L}_{\mathsf{IB}}(\theta; X_i, Y_i)$
---

space $\mathcal{Z}$. Let $c_\theta : \mathcal{Z} \times \mathcal{Z} \rightarrow \mathbb{R}_+ \cup \{\infty\}$ denote the transportation cost of moving mass from $(x_0, y_0)$ to $(x, y)$: $c_\theta((x_0, y_0), (x, y)) := \|z_0 - z\|_2^2 + \infty \cdot \mathbf{1}\{y_0 \neq y\}$, where $z = f(\theta_f; x)$. For probability measures $P$ and $P_0$ supported on $\mathcal{Z}$, let $\Pi(P, P_0)$ be their couplings. Then, we use the *Wasserstein metric* $D_\theta$ defined by $D_\theta(P, P_0) := \inf_{M \in \Pi(P, P_0)} \{\mathbb{E}_M [c_\theta((X_0, Y_0), (X, Y))]\}$. The solution to the worst-case problem (2) ensures good performance (robustness) against any data distribution $P$ that is $\rho$ distance away from the source domain $P_0$. However, for deep neural networks, this formulation is intractable with arbitrary $\rho$. Instead, following the reformulation of [51, 65], we consider its *Lagrangian relaxation* $\mathcal{F}$ for a fixed penalty parameter $\gamma \geq 0$:

$$\underset{\theta \in \Theta}{\text{minimize}} \left\{ \mathcal{F}(\theta) := \sup_P \{\mathbb{E}[\mathcal{L}(\theta; X, Y)] - \gamma D_\theta(P, P_0)\} \right\}. \tag{4}$$

## 3 Methodology

In this paper, our main idea is to incorporate the IB principle into adversarial data augmentation so as to improve model robustness to large domain shifts. We start by adapting the IB Lagrangian (1) to supervised-learning scenarios so that the latent representation $Z$ can be leveraged for classification purposes. To this end, we modify the IB Lagrangian (1) following [1, 2, 5] to $\mathcal{L}_{\mathsf{IB}}(\theta; X, Y) := \mathcal{L}_{\mathsf{CE}}(\theta; X, Y) + \beta I(X; Z)$, where the constraint on $I(Y; Z)$ is replaced with the risk associated to the prediction according to the loss function $\mathcal{L}_{\mathsf{CE}}$. We can see that $\mathcal{L}_{\mathsf{IB}}$ appears as a standard cross-entropy loss augmented with a regularizer $I(X; Z)$ promoting minimality of the representation. Then, we rewrite Eq. (4) to leverage the newly defined $\mathcal{L}_{\mathsf{IB}}$ loss function:

$$\underset{\theta \in \Theta}{\text{minimize}} \left\{ \mathcal{F}_{\mathsf{IB}}(\theta) := \sup_P \{\mathbb{E}[\mathcal{L}_{\mathsf{IB}}(\theta; X, Y)] - \gamma D_\theta(P, P_0)\} \right\}. \tag{5}$$

As discussed in [51, 65], the worst-case setting of Eq. (5) can be formalized as a *minimax* optimization problem. It is solved by an iterative training procedure where two phases are alternated in $K$ iterations. In the *maximization* phase, new data points are produced by computing the inner maximization problem $\mathcal{F}_{\mathsf{IB}}$ to mimic fictitious target distributions $P$ that satisfy the constraint $D_\theta(P, P_0) \leq \rho$. In the *minimization* phase, the network parameters are updated by the loss function $\mathcal{L}_{\mathsf{IB}}$ evaluated on the adversarial examples generated from the maximization phase.

The main challenge in optimizing Eq. (5) is that exact computation of the compression term $I(X; Z)$ in $\mathcal{L}_{\mathsf{IB}}$ is almost impossible due to the high dimensionality of the data. The way of approximating this term in the minimization phase has been widely studied in recent years, and we follow [17, 19] to express $I(X; Z)$ by $\ell_2$ penalty (also known as weight decay [33]). Below, we discuss how to effectively implement $I(X; Z)$ in the maximization phase for adversarial data augmentation. The full algorithm is summarized in Algorithm 1.

## 3.1 Regularizing Maximization Phase via Maximum Entropy

Intuitively, regularizing the mutual information $I(X;Z)$ in the maximization phase encourages adversarial perturbations that cannot be effectively "compressed" by the current model. From the information theory perspective, these perturbations usually imply large domain shifts, and thus can potentially benefit model generalization and robustness. However, since $Z$ is high dimensional, maximizing $I(X;Z)$ is intractable. One of our key results is that, when we restrict to classification scenarios, we can efficiently approximate and maximize $I(X;Z)$ during adversarial data augmentation. As we will show, this process can be effectively implemented through maximizing the entropy of network predictions, which is a tractable lower bound of $I(X;Z)$.

To set the stage, we let $\hat{Y} \in \mathcal{Y}$ denote the predicted class label given the input $X$. As described in [5, 58], deep neural networks can be considered as a Markov chain of successive representations of the input where information flows obeying the structure: $X \to Z \to \hat{Y}$. By the *Data Processing Inequality* [12], we have $I(X;Z) \geq I(X;\hat{Y})$. On the other hand, when performing data augmentation during each maximization phase, the model parameters $\theta$ are fixed; and $\hat{Y}$ is a *deterministic* function of $X$, i.e., any given input is mapped to a single class. Consequently, it holds that $H(\hat{Y}|X) = 0$, where $H(\cdot)$ is the *Shannon entropy*. After combining all these together, then we have Proposition 1.

**Proposition 1.** *Consider a deterministic neural network, the parameters $\theta$ of which are fixed. Given the input $X$, let $\hat{Y}$ be the network prediction and $Z$ be the latent representation of $X$. Then, the mutual information $I(X;Z)$ is lower bounded by $H(\hat{Y})$, i.e., we have that,*

$$I(X;Z) \geq I(X;\hat{Y}) = H(\hat{Y}) - H(\hat{Y}|X) = H(\hat{Y}). \tag{6}$$

Note that Eq. (6) does not mean that calculating $H(\hat{Y})$ does not need $X$, since $\hat{Y}$ is generated by inputting $X$ into the network. There are two important benefits of our formulation (6) to be discussed. First, it provides a method to maximize $I(X;Z)$ that is not related to the input dimensionality. Thus, for high dimensional images, we can still maximize mutual information this way. Second, our formulation is closely related to the *Deterministic Information Bottleneck* [56], where $I(X;Z)$ is approximated by $H(Z)$. However, $H(Z)$ is still intractable in general. Instead, $H(\hat{Y})$ can be directly computed from the softmax output of a classification network as we will show later. Next, we modify Eq. (5) by replacing $I(X;Z)$ with $H(\hat{Y})$, which becomes a relaxed worst-case problem:

$$\mathcal{F}_{\mathsf{DIB}}(\theta) := \sup_{P} \left\{ \mathbb{E}[\mathcal{L}_{\mathsf{CE}}(\theta; X, Y) + \beta H(\hat{Y})] - \gamma D_{\theta}(P, P_0) \right\}. \tag{7}$$

From a Bayesian perspective, the prediction entropy $H(\hat{Y})$ can be viewed as the predictive uncertainty [28, 52] of a model. Therefore, our maximum-entropy formulation (7) is equivalent to perturbing the underlying data distribution so that the predictive uncertainty of the current model is enlarged in the maximization phase. This motivates us to extend our approach to *stochastic* neural networks for better capturing the model uncertainty as we will show in the experiment.

**Empirical Estimation.** Now, $\mathcal{F}_{\mathsf{DIB}}$ involves the expected prediction entropy $H(\hat{Y})$ over the data distribution. However, during training we only have sample access to the data distribution, which we can use as a surrogate for empirical estimation. Given an observed input $x$ sampled from the source distribution $P_0$, we start from defining the prediction entropy of its corresponding output by:

$$h(\theta; x) := -\sum_{i=1}^{|\mathcal{Y}|} p_{(i)}(\theta; x) \log p_{(i)}(\theta; x). \tag{8}$$

Then, through calculating the expectation over the prediction entropies of all possible observations $\{x_i\}_{1 \leq i \leq N}$ contained in the source dataset $P_0$, we can obtain the empirical estimation of $H(\hat{Y})$:

$$\hat{H}(\hat{Y}) := -\sum_{\mathcal{Y}} \left( \sum_i p_{\hat{Y}|X}(\cdot|x_i)\hat{p}(x_i) \right) \log \left( \sum_i p_{\hat{Y}|X}(\cdot|x_i)\hat{p}(x_i) \right) \approx \frac{1}{N}\sum_{i=1}^{N} h(\theta; x_i), \tag{9}$$

where $\hat{H}(\cdot)$ denotes the empirical entropy, $\hat{p}$ is the empirical distribution of $x_i$, and the approximation is achieved by Jensen's inequality. After combing Eq. (9) with the relaxed worst-case

problem (7), we will have the empirical counterpart of $\mathcal{F}_{\text{DIB}}$ which is defined by $\hat{\mathcal{F}}_{\text{DIB}}(\theta) :=$ $\sup_P \{\mathbb{E}_{x\sim P}[\mathcal{L}_{\text{CE}}(\theta; x, y) + \beta h(\theta; x)] - \gamma D_\theta(P, P_0)\}$. Taking the dual reformulation of the penalty problem $\hat{\mathcal{F}}_{\text{DIB}}$, we can obtain an efficient solution procedure. The following result is a minor adaptation of [65] (Lemma 1):

**Proposition 2.** *Let $\mathcal{L} : \Theta \times (\mathcal{X} \times \mathcal{Y}) \to \mathbb{R}$ and $h : \Theta \times \mathcal{X} \to \mathbb{R}$ be continuous. Let $\phi_\gamma$ denote the robust surrogate loss. Then, for any distribution $P_0$ and any $\gamma \geq 0$, we have that,*

$$\sup_P \{\mathbb{E}_{x\sim P}[\mathcal{L}(\theta; x, y) + \beta h(\theta; x)] - \gamma D_\theta(P, P_0)\} = \mathbb{E}_{x\sim P_0}\left[\phi_\gamma(\theta; x, y)\right], \quad (10)$$

$$where \ \phi_\gamma(\theta; x_0, y_0) := \sup_{x\in\mathcal{X}} \{\mathcal{L}(\theta; x, y_0) + \beta h(\theta; x) - \gamma c_\theta((x, y_0), (x_0, y_0))\}. \quad (11)$$

To solve the penalty problem of Eq. (5), in the minimization phase of the iterative training procedure, we can perform *Stochastic Gradient Descent* (SGD) on the robust surrogate loss $\phi_\gamma$. To be specific, under suitable conditions [10], we have that $\nabla_\theta \phi_\gamma(\theta; x_0, y_0) = \nabla_\theta \mathcal{L}_{\text{IB}}(\theta; x_\gamma^\star, y_0)$, where $x_\gamma^\star = \operatorname{argmax}_{x\in\mathcal{X}} \{\mathcal{L}_{\text{CE}}(\theta; x, y_0) + \beta h(\theta; x) - \gamma c_\theta((x, y_0), (x_0, y_0))\}$ is an adversarial perturbation of $x_0$ at the current model $\theta$. On the other hand, in the maximization phase, we solve the maximization problem (11) by *Maximum-Entropy Adversarial Data Augmentation* (ME-ADA) in this work. Concretely, in the $k$-th *maximization* phase, we compute $N$ adversarially perturbed samples at the current model $\theta$:

$$X_i^k \in \operatorname*{argmax}_{x\in\mathcal{X}} \left\{\mathcal{L}_{\text{CE}}(\theta; x, Y_i) + \beta h(\theta; x) - \gamma c_\theta((x, Y_i), (X_i^{k-1}, Y_i))\right\}. \quad (12)$$

Note that the entropy term $h(\theta; x)$ is efficient to be calculated from the softmax output of a model, which can be implemented with one line of code in modern deep learning frameworks, and substantial performance improvement can be achieved by it as we will show in the experiments.

**Theoretic Bound.** It is essential to guarantee that the empirical estimate of the entropy $\hat{H}$ (from a training set $\mathcal{S}$ containing $N$ samples) is an accurate estimate of the true expected entropy $H$. The next proposition ensures that for large $N$, in a classification problem, the sample estimate of average entropy is close to the expected entropy.

**Proposition 3.** *Let $Y$ be a fixed probabilistic function of $X$ into an arbitrary finite target space $\mathcal{Y}$, determined by a fixed and known conditional probability distribution $p_{Y|X}$, and $\mathcal{S}$ be a sample set of size $N$ drawn from the joint probability distribution $p_{XY}$. For any $\delta \in (0, 1)$, with probability of at least $1 - \delta$ over the sample set $\mathcal{S}$, we have,*

$$|H(Y) - \hat{H}(Y)| \leq \frac{|\mathcal{Y}|\log(N)\sqrt{\log(2/\delta)}}{\sqrt{2N}} + \frac{|\mathcal{Y}| - 1}{N}. \quad (13)$$

We prove Proposition 3 in the supplementary material. The proof adapts the setting in [50], where we bound the deviations of the information estimations from their expectation and then use the bound on the expected bias of entropy estimation. Here, it is also worth discussing two important properties of this bound. First, we note that Proposition 3 holds for any fixed probabilistic function. Compared with prior studies on the plug-in estimate of discrete entropy over a finite size alphabet [62, 68], we focus on the bound of non-optimal estimators. In particular, this proposition holds for any $\hat{Y}$, even if $\hat{Y}$ is not a globally optimal solution for $\mathcal{F}_{\text{DIB}}$ in Eq. (7). This is the case of models in the maximization phase, which thus ensures the effectiveness of our formulation across the whole iterative training procedure. Second, the bound does not depend on $|\mathcal{X}|$. In addition, the complexity of the bound is mainly controlled by $|\mathcal{Y}|$. By constraining $|\mathcal{Y}|$ to be small, a tight bound can be achieved. This assumption usually holds for the setting of training classification models, i.e., $|\mathcal{Y}| \ll N$.

## 3.2 Maximum Entropy in Non-Deterministic Conditions

It is important to note that not all models are deterministic, e.g., when deep neural networks are stochastic [18, 57] or contain Dropout layers [20, 55]. The mapping from $X$ to $\hat{Y}$ may be intrinsically noisy or non-deterministic. Here, we show that when $\hat{Y}$ is a small perturbation away from being a deterministic function of $X$, our maximum-entropy formulation (7) still applies in an approximate sense. We now consider the case when the joint distribution of $X$ and $\hat{Y}$ is $\epsilon$-close to having $\hat{Y}$ be a deterministic function of $X$. The next result is a minor adaptation of [30] (Theorem 1) and it shows that the conditional entropy $H(\hat{Y}|X)$ is $\mathcal{O}(-\epsilon \log \epsilon)$ away from being zero.

**Corollary 1.** *Let $X$ be a random variable and $Y$ be a random variable with a finite set of outcomes $\mathcal{Y}$. Let $p_{XY}$ be a joint distribution over $X$ and $Y$ under which $Y = f(X)$. Let $\tilde{p}_{XY}$ be a joint distribution over $X$ and $Y$ which has the same marginal over $X$ as $p_{XY}$, i.e., $\tilde{p}_X = p_X$, and obey $|\tilde{p}_{XY} - p_{XY}|_1 \leq \epsilon \leq \frac{1}{2}$. Then, we have that,*

$$|H(\tilde{p}(Y|X))| \leq -\epsilon \log \frac{\epsilon}{|\mathcal{Y}|^3}. \tag{14}$$

As we show in this corollary, even if the relationship between $X$ and $\hat{Y}$ is not perfectly deterministic but close to being so, i.e., it is $\epsilon$-close to a deterministic function, then we have $H(\hat{Y}|X) \approx 0$. Hence, in this case, the proposed Proposition 1 and our maximum-entropy adversarial data augmentation formulation (7) still hold in an approximate sense.

## 4 Experiments

In this section, we evaluate our approach over a variety of settings. We first test with MNIST under the setting of large domain shifts, and then test on a more challenging dataset, with PACS data under the domain generalization setting. Further, we test on CIFAR-10-C and CIFAR-100-C which are standard benchmarks for evaluating model robustness to common corruptions. We compare the proposed *Maximum-Entropy Adversarial Data Augmentation* (ME-ADA) with previous state of the art when available. We note that *Adversarial Data Augmentation* (ADA) [65] is our main competitor, since our method downgrades to [65] when the maximum-entropy term is discarded.

**Datasets.** MNIST dataset [38] consists of handwritten digits with 60,000 training examples and 10,000 testing examples. Other digit datasets, including SVHN [45], MNIST-M [21], SYN [21] and USPS [14], are leveraged for evaluating model performance. These four datasets contain large domain shifts from MNIST in terms of backgrounds, shapes and textures. PACS [39] is a recent dataset with different object style depictions and a more challenging domain shift than the MNIST experiment. This dataset contains four domains (art, cartoon, photo and sketch), and shares seven common object categories (dog, elephant, giraffe, guitar, house, horse and person) across these domains. It is made up of 9,991 images with the resolution of $224 \times 224$. For fair comparison, we follow the protocol in [39] including the recommended train, validation and test split.

CIFAR-10 and CIFAR-100 are two datasets [31] containing small $32 \times 32$ natural RGB images, both with 50,000 training images and 10,000 testing images. CIFAR-10 has 10 categories, and CIFAR-100 has 100 object classes. In order to measure the resilience of a model to common corruptions, we evaluate on CIFAR-10-C and CIFAR-100-C datasets [24]. These two datasets are constructed by corrupting the original CIFAR test sets. For each dataset, there are a total of fifteen noise, including blur, weather, and digital corruption types, and each of them appears at five severity levels or intensities. We do not tune on the validation corruptions, so we report the average performance over all corruptions and intensities.

### 4.1 MNIST with Domain Shifts

**Experiment Setup.** We follow the setup of [65] in experimenting with MNIST dataset. We use 10,000 samples from MNIST for training and evaluate prediction accuracy on the respective test sets of SVHN, MNIST-M, SYN and USPS. In order to work with comparable datasets, we resize all the images to $32 \times 32$, and treat images from MNIST and USPS as RGB images. We use LeNet [37] as a base model and the batch size is 32. We use Adam [29] with $\alpha = 0.0001$ for minimization and SGD with $\eta = 1.0$ for maximization. We set $T_{\mathsf{MIN}} = 100$, $T_{\mathsf{MAX}} = 15$, $\gamma = 1.0$, $\beta = 10.0$ and $K = 3$. We compare our method against ERM [63], ADA [65], and PAR [66].

We also implement a variant of our method through *Bayesian Neural Networks* (BNNs) [8, 20, 36] to demonstrate our compatibility with stochastic neural networks. BNNs learn a distribution over network parameters and are currently the state of the art for estimating predictive uncertainty [16, 44]. We follow [8] to implement the BNN via variational inference. During the training procedure, in each maximization phase, a set of network parameters are drawn from the variational posterior $q(\theta|\cdot)$, and then the predictive uncertainty is redefined by the expectation of all prediction entropies: $\bar{h}(q; x) = \mathbb{E}_q[h(\theta; x)]$. We refer to the supplementary material for more details of this BNN variant.

Table 1: Average classification accuracy (%) and standard deviation of models trained on MNIST [38] and evaluated on SVHN [45], MNIST-M [21], SYN [21] and USPS [14]. The results are averaged over ten runs. Best performances are highlighted in bold. The results of PAR are obtained from [70].

| | SVHN [45] | MNIST-M [21] | SYN [21] | USPS [14] | Average |
|---|---|---|---|---|---|
| Standard (ERM [63]) | $31.95 \pm 1.91$ | $55.96 \pm 1.39$ | $43.85 \pm 1.27$ | $79.92 \pm 0.98$ | $52.92 \pm 0.98$ |
| PAR [66] | $36.08 \pm 1.27$ | $61.16 \pm 0.21$ | $45.48 \pm 0.35$ | $79.95 \pm 1.18$ | $55.67 \pm 0.33$ |
| Adv. Augment (ADA) [65] | $35.70 \pm 2.00$ | $58.65 \pm 1.72$ | $47.18 \pm 0.61$ | $80.40 \pm 1.70$ | $55.48 \pm 0.74$ |
| + *Max Entropy* (ME-ADA) | $42.00 \pm 1.74$ | $\mathbf{63.98} \pm 1.82$ | $49.80 \pm 1.74$ | $79.10 \pm 1.03$ | $58.72 \pm 1.12$ |
| + *Max Entropy* w/ BNN | $\mathbf{42.56} \pm 1.45$ | $63.27 \pm 2.09$ | $\mathbf{50.39} \pm 1.29$ | $\mathbf{81.04} \pm 0.98$ | $\mathbf{59.32} \pm 0.82$ |

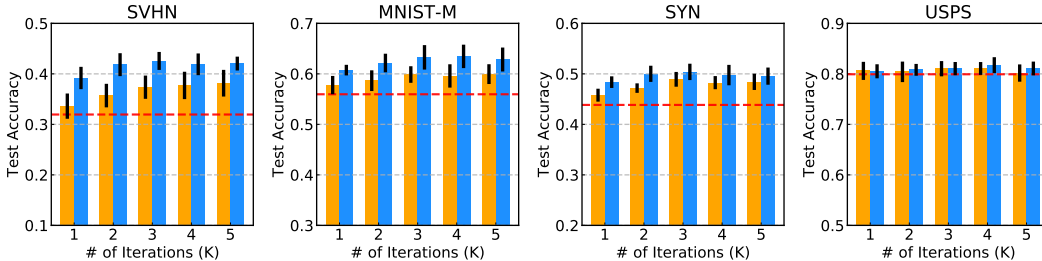

Figure 1: Test accuracy associated with models trained using 10,000 MNIST [38] samples and tested on SVHN [45], MNIST-M [21], SYN [21] and USPS [14]. We compare our method (*blue*) to [65] (*orange*) with different number of iterations $K$, and Empirical Risk Minimization (ERM) [63] (*red line*). The results are averaged over ten runs; and black bars indicate the range of accuracy spanned.

**Results.** Table 1 shows the classification accuracy and standard deviation of each model averaged over ten runs. We can see that our model with the maximum-entropy formulation achieves the best performance, while the improvement on USPS is not as significant as those on other domains due to its high similarity with MNIST. We then notice that, after engaging the BNN, our performance is further improved. Intuitively, we believe this is because the BNN provides a better estimation of the predictive uncertainty in the maximization phase. We are also interested in analyzing the behavior of our method when $K$ is increased. Figure 1 shows the results of our method and other baselines by varying the number of iterations $K$ while fixing $\gamma$ and $\beta$. We observe that our method improves performances on SVHN, MNIST-M and SYN, outperforming both ERM and [65] statistically significantly in different iterations. This demonstrates that the improvements obtained by our method are consistent.

## 4.2 PACS

**Experiment Setup.** We continue to experiment on PACS dataset, which consists of collections of images over four domains. Each time, one domain is selected as the test domain, and the rest three are used for training. Following [39], we use the ImageNet pretrained AlexNet [32] as a base network. We compare with recently reported state of the art engaging domain identifications, including DSN [9], L-CNN [39], MLDG [40], Fusion [43], MetaReg [6] and Epi-FCR [41], as well as methods forgoing domain identifications, including AGG [41], HEX [67], and PAR [66]. Former methods often obtain better results because they utilize domain identifications. Our method belongs to the latter category. Other training details are provided in the supplementary material.

**Results.** We report the results in Table 2. We note that our method achieves the best performance among techniques forgoing domain identifications. More impressively, our method, without using domain identifications, is only slightly shy of MetaReg [6] in terms of overall performance, which takes advantage domain identifications. Interestingly, it is also worth mentioning that our method improves previous methods with a relatively large margin when "sketch" is the testing domain. This is notable because "sketch" is the only colorless domain which owns the largest domain shift out of the four domains in PACS. Our method handles this extreme case by producing larger data shifts from the source domain with the proposed maximum-entropy term during data augmentation.

Table 2: Classification accuracy (%) of our approach on PACS dataset [39] in comparison with the previously reported state-of-the-art results. Bold numbers indicate the best performance (two sets, one for each scenario engaging or forgoing domain identifications, respectively).

| | DSN | L-CNN | MLDG | Fusion | MetaReg | Epi-FCR | AGG | HEX | PAR | ADA | ME-ADA |
|---|---|---|---|---|---|---|---|---|---|---|---|
| Domain ID | ✓ | ✓ | ✓ | ✓ | ✓ | ✓ | ✗ | ✗ | ✗ | ✗ | ✗ |
| Art | 61.1 | 62.9 | 66.2 | 64.1 | **69.8** | 64.7 | 63.4 | 66.8 | 66.9 | 64.3 | **67.1** |
| Cartoon | 66.5 | 67.0 | 66.9 | 66.8 | 70.4 | **72.3** | 66.1 | 69.7 | 67.1 | 69.8 | **69.9** |
| Photo | 83.3 | 89.5 | 88.0 | 90.2 | **91.1** | 86.1 | 88.5 | 87.9 | **88.6** | 85.1 | **88.6** |
| Sketch | 58.6 | 57.5 | 59.0 | 60.1 | 59.2 | **65.0** | 56.6 | 56.3 | 62.6 | 60.4 | **63.0** |
| Average | 67.4 | 69.2 | 70.0 | 70.3 | **72.6** | 72.0 | 68.7 | 70.2 | 71.3 | 69.9 | **72.2** |

Table 3: Average classification accuracy (%). Across several architectures, our approach obtains CIFAR-10-C and CIFAR-100-C corruption robustness that exceeds the previous state of the art by a large margin. Best performances are highlighted in bold.

| | | Standard | Cutout | CutMix | AutoDA | Mixup | AdvTrain | ADA | ME-ADA |
|---|---|---|---|---|---|---|---|---|---|
| CIFAR-10-C | AllConvNet | 69.2 | 67.1 | 68.7 | 70.8 | 75.4 | 71.9 | 73.0 | **78.2** |
| | DenseNet | 69.3 | 67.9 | 66.5 | 73.4 | 75.4 | 72.4 | 69.8 | **76.9** |
| | WideResNet | 73.1 | 73.2 | 72.9 | 76.1 | 77.7 | 73.8 | 79.7 | **83.3** |
| | ResNeXt | 72.5 | 71.1 | 70.5 | 75.8 | 77.4 | 73.0 | 78.0 | **83.4** |
| Average | | 71.0 | 69.8 | 69.7 | 74.0 | 76.5 | 72.8 | 75.1 | **80.5** |
| CIFAR-100-C | AllConvNet | 43.6 | 43.2 | 44.0 | 44.9 | 46.6 | 44.0 | 45.3 | **51.2** |
| | DenseNet | 40.7 | 40.4 | 40.8 | 46.1 | 44.6 | 44.8 | 45.2 | **47.8** |
| | WideResNet | 46.7 | 46.5 | 47.1 | 50.4 | 49.6 | 44.9 | 50.4 | **52.8** |
| | ResNeXt | 46.6 | 45.4 | 45.9 | 48.7 | 48.6 | 45.6 | 53.4 | **57.3** |
| Average | | 44.4 | 43.9 | 44.5 | 47.5 | 47.4 | 44.8 | 48.6 | **52.3** |

## 4.3 CIFAR-10 and CIFAR-100 with Corruptions

**Experiment Setup.** In the following experiments, we show that our approach endows robustness to various architectures including All Convolutional Network (AllConvNet) [49, 54], DenseNet-BC [26] (with $k = 12$ and $d = 100$), WideResNet (40-2) [73], and ResNeXt-29 ($32 \times 4$) [69]. We train all networks with an initial learning rate of 0.1 optimized by SGD using Nesterov momentum, and the learning rate decays following a cosine annealing schedule [42]. All input images are pre-processed with standard random left-right flipping and cropping in the minimization phase. We train AllConvNet and WideResNet for 100 epochs; DenseNet and ResNeXt require 200 epochs for convergence. Following the setting of [25], we use a weight decay of 0.0001 for DenseNet and 0.0005 otherwise. Due to the space limitation, we ask the readers to refer to the supplementary material for detailed settings of our training parameters for different architectures.

**Baselines.** To demonstrate the utility of our approach, we compare to many state-of-the-art techniques designed for robustness to image corruptions. These baseline techniques include (i) the standard data augmentation baseline and Mixup [74]; (ii) two regional regularization strategies for images, i.e., Cutout [15] and Cutmix [72]; (iii) AutoAugment [13], which searches over data augmentation policies to find a high-performing data augmentation policy via reinforcement learning; (iv) Adversarial Training [27] for model robustness against unforeseen adversaries, and Adversarial Data Augmentation [65] which generates adversarial perturbations using Wasserstein distances.

**Results.** The results are shown in Table 3. Our method enjoys the best performance and improves previous state of the art by a large margin (5% of accuracy on CIFAR-10-C and 4% on CIFAR-100-C). More importantly, these gains are achieved across different architectures and on both datasets. Figure 2 shows more detailed comparisons over all corruptions. We find that our substantial gains in robustness are spread across a wide variety of corruptions, with a small drop of performance in only three corruption types: fog, brightness and contrast. Especially, for glass blur, Gaussian, shot and impulse noises, accuracies are significantly improved by 25%. From the Fourier perspective [71], the performance gains from our adversarial perturbations lie primarily in high frequency domains, which are commonly occurring image corruptions. These results demonstrate that the maximum-entropy term can regularize networks to be more robust to common image corruptions.

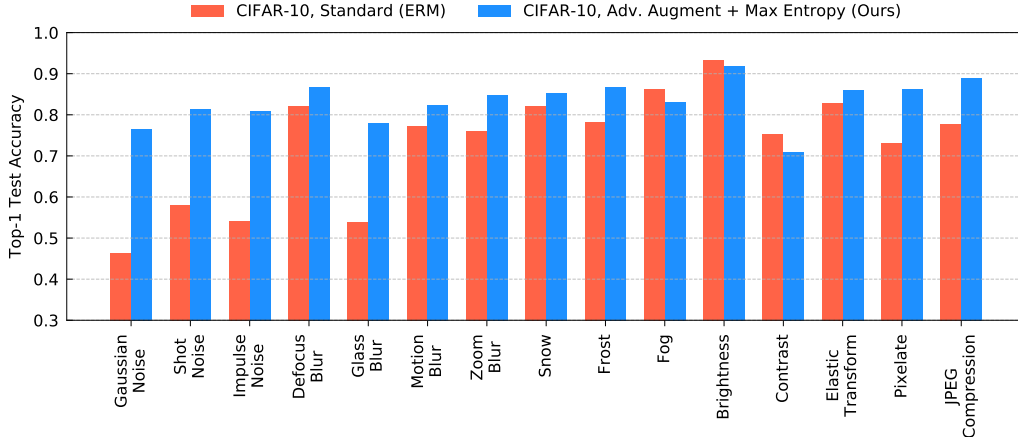

Figure 2: Test accuracy of the Empirical Risk Minimization (ERM) [63] principle compared to our approach on the fifteen CIFAR-10-C [24] corruptions using WideResNet (40-2) [73]. Each bar represents an average over all five corruption strengths for a given corruption type.

## 5   Conclusion

In this work, we introduced a *maximum-entropy* technique that regularizes adversarial data augmentation. It encourages the model to learn with fictitious target distributions by producing "hard" adversarial perturbations that enlarge predictive uncertainty of the current model. As a result, the learned model is able to achieve improved robustness to large domain shifts or corruptions encountered during deployment. We demonstrate that our technique obtains state-of-the-art performance on MNIST, PACS, and CIFAR-10/100-C, and is extremely simple to implement. One major limitation of our method is that it cannot be directly applied to regression problems since the maximum-entropy lower bound is still difficult to compute in this case. Our future work might consider alternative measurements of information [46, 60] that are more suited for general machine learning applications.

## Broader Impact

The proposed method will be used to train a perception system that can robustly and reliably classify object instances. For example, this system can be used in many fundamental real-world applications in which a user desires to classify object instances from a product database, such as products found on local supermarkets or online stores. Similar to most deep learning applications learning from data which run the risk of producing biased or offensive content reflecting the training data, our work that learns a data-driven classification model is no exception. Our method moderates this issue by producing efficient fictitious target domains that are largely shifted from the source training dataset, so that the trained model on these adversarial domains are less biased. However, a downside of this moderation is the introduction of new hyper-parameters to be tuned for different tasks. Compared with other methods that obtain the same robustness but have to be trained on larger datasets, the proposed research can significantly reduce the data collection from different domains to train classification models, thereby reducing the system development time and lower related costs.

## Acknowledgments and Disclosure of Funding

This research was funded based on partial funding to Dimitris Metaxas from NSF: IIS 1703883, CNS-1747778, CCF-1733843, IIS-1763523, IIS-1849238-825536 and MURI-Z8424104-440149.

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
