[Supplementary Material]

# Maximum-Entropy Adversarial Data Augmentation for Improved Generalization and Robustness: Supplementary Material

**Long Zhao**[1]    **Ting Liu**[2]    **Xi Peng**[3]    **Dimitris Metaxas**[1]
[1]Rutgers University    [2]Google Research    [3]University of Delaware
{lz311,dnm}@cs.rutgers.edu    liuti@google.com    xipeng@udel.edu

## 1  Proofs

### 1.1  Proof of Proposition 3

Here, we follow the guidance of [15] to prove Proposition 3. Let $\mathcal{S}$ be a sample set of size $m$, and let $T$ be a probabilistic function of $X$ into an arbitrary finite target space, defined by $p(t|x)$ for all $x \in \mathcal{X}$ and $t \in \mathcal{T}$. To prove Proposition 3, we bound the deviations of the entropy estimations from its expectation: $|\hat{H}(T)| - \mathbb{E}[\hat{H}(T)]|$, and then use a bound on the expected bias of entropy estimation.

To bound the deviation of the entropy estimates, we use McDiarmid's inequality [13], in a manner similar to [1]. For this, we must bound the change in value of each of the entropy estimations when a single instance in $\mathcal{S}$ is arbitrarily changed. A useful and easily proven inequality in that regard is the following: for any natural $m$ and for any $a \in [0, 1 - 1/m]$ and $\Delta \le 1/m$,

$$|(a + \Delta)\log(a + \Delta) - a\log(a)| \le \frac{\log(m)}{m}. \tag{1}$$

With this in equality, a careful application of McDiarmid's inequality leads to the following lemma.

**Lemma 1.** *For any $\delta \in (0,1)$, with probability of at least $1 - \delta$ over the sample set, we have that,*

$$|\hat{H}(T) - \mathbb{E}[\hat{H}(T)]| \le \frac{|\mathcal{T}|\log(m)\sqrt{\log(2/\delta)}}{\sqrt{2m}}. \tag{2}$$

*Proof.* First, we bound the change caused by a single replacement in $\hat{H}(T)$. We have that,

$$\hat{H}(T) = -\sum_t \left(\sum_x p(t|x)\hat{p}(x)\right) \log\left(\sum_x p(t|x)\hat{p}(x)\right). \tag{3}$$

If we change a single instance in $\mathcal{S}$, then there exist two pairs $(x, y)$ and $(x', y')$ such that $\hat{p}(x, y)$ increases by $1/m$, and $\hat{p}(x', y')$ decreases by $1/m$. This means that $\hat{p}(x)$ and $\hat{p}(x')$ also change by at most $1/m$, while all other values in the distribution remain the same. Therefore, for each $t \in \mathcal{T}$, $\sum_x p(t|x)\hat{p}(x)$ changes by at most $1/m$.

Based on this and Eq. (1), $\hat{H}(T)$ changes by at most $|\mathcal{T}|\log(m)/m$. Applying McDiarmid's inequality, we get Eq. (2). We have thus proven Lemma 1. □

Lemma 1 provides bounds on the deviation of the $\hat{H}(T)$ from their expected values. In order to relate these to the true values of the entropy $H(T)$, we use the following bias bound from [14] and [15].

**Lemma 2** (Paninski [14]; Shamir et al. [15], Lemma 9). *For a random variable $T \in \mathcal{T}$, with the plug-in estimation $\hat{H}(\cdot)$ on its entropy, based on an i.i.d. sample set of size $m$, we have that,*

$$|\mathbb{E}[H(T) - \hat{H}(T)]| \leq \log\left(1 + \frac{|\mathcal{T}| - 1}{m}\right) \leq \frac{|\mathcal{T}| - 1}{m}. \tag{4}$$

From this lemma, the quantity $|\mathbb{E}[H(T) - \hat{H}(T)]|$ is upper bounded by $(|\mathcal{T}| - 1)/m$. Combining it with Eq. (2), we get the bound in Proposition 3.

**Proposition 3.** *Let $Y$ be a fixed probabilistic function of $X$ into an arbitrary finite target space $\mathcal{Y}$, determined by a fixed and known conditional probability distribution $p_{Y|X}$, and $\mathcal{S}$ be a sample set of size $N$ drawn from the joint probability distribution $p_{XY}$. For any $\delta \in (0,1)$, with probability of at least $1 - \delta$ over the sample set $\mathcal{S}$, we have,*

$$|H(Y) - \hat{H}(Y)| \leq \frac{|\mathcal{Y}|\log(N)\sqrt{\log(2/\delta)}}{\sqrt{2N}} + \frac{|\mathcal{Y}| - 1}{N}. \tag{5}$$

*Proof.* To prove the proposition, we start by using the triangular inequality to write,

$$|H(Y) - \hat{H}(Y)| \leq |H(Y) - \mathbb{E}[\hat{H}(Y)]| + |\hat{H}(Y) - \mathbb{E}[\hat{H}(Y)]|. \tag{6}$$

Because $H(Y)$ is constant, we have:

$$|H(Y) - \mathbb{E}[\hat{H}(Y)]| = |\mathbb{E}[H(Y)] - \mathbb{E}[\hat{H}(Y)]|. \tag{7}$$

By the linearity of expectation, we have:

$$|\mathbb{E}[H(Y)] - \mathbb{E}[\hat{H}(Y)]| = |\mathbb{E}[H(Y) - \hat{H}(Y)]|. \tag{8}$$

Combining these with Lemmas 1 and 2, we get the bound in Proposition 3. □

## 1.2 Proof of Corollary 1

The proof of Corollary 1 is based on the following bound proposed by [7].

**Lemma 3** (Kolchinsky et al. [7], Theorem 1). *Let $Z$ be a random variable (continuous or discrete), and $Y$ be a random variable with a finite set of outcomes $\mathcal{Y}$. Consider two joint distributions over $Z$ and $Y$, $p_{ZY}$ and $\tilde{p}_{ZY}$, which have the same marginal over $Z$, $p(z) = \tilde{p}(z)$, and obey $|p_{ZY} - \tilde{p}_{ZY}|_1 \leq \epsilon \leq \frac{1}{2}$. Then,*

$$|H(p(Y|X)) - H(\tilde{p}(Y|X))| \leq -\epsilon \log \frac{\epsilon}{|\mathcal{Y}|^3}. \tag{9}$$

This lemma upper bounds the quantity $|H(p(Y|X)) - H(\tilde{p}(Y|X))|$ by $-\epsilon \log\log(\epsilon/|\mathcal{Y}|^3)$. After extending it to the case when $Y$ is a deterministic function of $X$, we get the bound in Corollary 1.

**Corollary 1.** *Let $X$ be a random variable and $Y$ be a random variable with a finite set of outcomes $\mathcal{Y}$. Let $p_{XY}$ be a joint distribution over $X$ and $Y$ under which $Y = f(X)$. Let $\tilde{p}_{XY}$ be a joint distribution over $X$ and $Y$ which has the same marginal over $X$ as $p_{XY}$, i.e., $\tilde{p}_X = p_X$, and obey $|\tilde{p}_{XY} - p_{XY}|_1 \leq \epsilon \leq \frac{1}{2}$. Then, we have that,*

$$|H(\tilde{p}(Y|X))| \leq -\epsilon \log \frac{\epsilon}{|\mathcal{Y}|^3}. \tag{10}$$

*Proof.* Since $Y$ is a deterministic function of $X$, i.e., $Y = f(X)$, we then have $H(p(Y|X)) = 0$. Combining this with Eq. (9), we prove the bound in Corollary 1. □

## 2 Implementation Details

### 2.1 BNN Variant

We follow [3] to implement the BNN variant of our method. Let $x$ be the observed input variable and $\theta$ be a set of latent variables. Deep neural networks can be viewed as a probabilistic model $p(y|x, \theta)$,

where $\mathcal{D} = \{x_i, y_i\}_i$ is a set of training examples and $y$ is the network output which belongs to a set of object categories by using the network parameters $\theta$. The variational inference aims to calculate this conditional probability distribution over the latent variables (network parameters) by finding the closest proxy to the exact posterior by solving an optimization problem.

Following the guidance of [3], we first assume a family of probability densities over the latent variables $\theta$ parameterized by $\psi$, i.e., $q(\theta|\psi)$. We then find the closest member of this family to the true conditional probability $p(\theta|\mathcal{D})$ by minimizing the KL-divergence between $q(\theta|\psi)$ and $p(\theta|\mathcal{D})$, which is equivalent to minimizing the following variational free energy:

$$\mathcal{L}_{\mathsf{BNN}}(\theta, \mathcal{D}) = \mathbb{D}_{\mathsf{KL}}\left[q(\theta|\psi)\|p(\theta)\right] - \mathbb{E}_{q(\theta|\psi)}\left[\log(p(\mathcal{D}|\theta))\right]. \tag{11}$$

This objective function can be approximated using $T$ Monte Carlo samples $\theta_i$ from the variational posterior [3]:

$$\mathcal{L}_{\mathsf{BNN}}(\theta, \mathcal{D}) \approx \sum_{i=1}^{T} \log(q(\theta_i|\psi)) - \log(p(\theta_i)) - \log(p(\mathcal{D}|\theta_i)). \tag{12}$$

We assume $q(\theta|\psi)$ have a Gaussian probability density function with diagonal covariance and parameterized by $\psi = (\mu, \sigma)$. A sample weight of the variational posterior can be obtained by the reparameterization trick [6]: we sample it from a unit Gaussian and parameterized by $\theta = \mu + \sigma \circ \epsilon$, where $\epsilon$ is the noise drawn from the unit Gaussian $\mathcal{N}(0, 1)$ and $\circ$ is the point-wise multiplication. For the prior, as suggested by [3], a scale mixture of two Gaussian probability density functions are chosen: they are zero-centered but have two different variances of $\sigma_1^2$ and $\sigma_2^2$ with the ratio of $\pi$. In this work, we let $-\log\sigma_1 = 0$, $-\log\sigma_2 = 6$, and $\pi = 0.25$. Then, the optimizing objective of adversarial perturbations in the maximization phase of our method is redefined by:

$$X_i^k \in \underset{x \in \mathcal{X}}{\operatorname{argmax}}\left\{\mathcal{L}_{\mathsf{CE}}(\theta; x, Y_i) + \beta \sum_{j=1}^{T} \frac{1}{T} h(\theta_j; x) - \gamma c_\theta((x, Y_i), (X_i^{k-1}, Y_i))\right\}, \tag{13}$$

where $\theta_j$ is sampled $T$ times from the learned variational posterior.

## 2.2 PACS

The learning principle of the previous state-of-the-art method on this dataset follows two streams. The first stream of methods, including DSN [4], L-CNN [9], MLDG [10], Fusion [12], MetaReg [2] and Epi-FCR [11], engages domain identifications, which means that when training the model, each source domain is regarded as a separate domain. The second stream of methods, containing AGG [11], HEX [17], and PAR [16], does not leverage domain identifications and combines all source domains into a single one during the training procedure. We can find that the first stream leverages more information, i.e., the domain identifications, during the network training, and thus often yields better performance than the second stream. Our work belongs to the latter stream.

Table 1: The settings of different target domains on PACS.

| Target Domain | $K$ | $T$ (loop) | $T_{\mathsf{MIN}}$ (loop) | $T_{\mathsf{MAX}}$ (loop) | $\alpha$ | $\eta$ | $\gamma$ | $\beta$ |
|---|---|---|---|---|---|---|---|---|
| Art | 1 | 45,000 | 100 | 50 | 0.001 | 50.0 | 10.0 | 1.0 |
| Cartoon | 1 | 45,000 | 100 | 50 | 0.001 | 50.0 | 10.0 | 100.0 |
| Photo | 1 | 45,000 | 100 | 50 | 0.001 | 50.0 | 10.0 | 1.0 |
| Sketch | 1 | 45,000 | 100 | 50 | 0.001 | 50.0 | 10.0 | 100.0 |

We follow the setup of [9] for network training. To align with the previous methods, the ImageNet pretrained AlexNet [8] is employed as the baseline network. In the network training, we set the batch size to 32. We use SGD with the learning rate of 0.001 (the learning rate decays following a cosine annealing schedule [18]), the momentum of 0.9, and weight decay of 0.00005 for minimization, while we use the SGD with the learning rate of 50.0 for maximization. Table 1 shows more detailed setting of all parameters under four different target domains.

Table 2: The settings of different network architectures on CIFAR-10-C and CIFAR-100-C.

| | | $K$ | $T$ (epoch) | $T_{\mathsf{MIN}}$ (epoch) | $T_{\mathsf{MAX}}$ (loop) | $\alpha$ | $\eta$ | $\gamma$ | $\beta$ |
|---|---|---|---|---|---|---|---|---|---|
| CIFAR-10-C | AllConvNet | 2 | 100 | 10 | 15 | 0.1 | 20.0 | 0.1 | 10.0 |
| | DenseNet | 2 | 200 | 10 | 15 | 0.1 | 20.0 | 1.0 | 100.0 |
| | WideResNet | 2 | 100 | 10 | 15 | 0.1 | 20.0 | 1.0 | 10.0 |
| | ResNeXt | 2 | 200 | 10 | 15 | 0.1 | 20.0 | 1.0 | 10.0 |
| CIFAR-100-C | AllConvNet | 2 | 100 | 10 | 15 | 0.1 | 20.0 | 0.1 | 10.0 |
| | DenseNet | 2 | 200 | 10 | 15 | 0.1 | 20.0 | 10.0 | 10.0 |
| | WideResNet | 2 | 100 | 10 | 15 | 0.1 | 20.0 | 1.0 | 10.0 |
| | ResNeXt | 2 | 200 | 10 | 15 | 0.1 | 20.0 | 10.0 | 10.0 |

## 2.3 CIFAR-10 and CIFAR-100

The experimental settings follow the setups in [5]. We use SGD for both minimization and maximization. In Table 2, we report the detailed settings of all parameters under different network architectures on CIFAR-10-C and CIFAR-100-C. Note that $T$ and $T_{\mathsf{MIN}}$ are measured by number of training epoches, while $T_{\mathsf{MAX}}$ is measured by number of iterations. In this work, we do not compare our method with [5], since the design of [5] depends on a set of pre-defined image corruptions which is with a different research target compared to our method.