[Reviews · NeurIPS 2020]

Review 1

Summary and Contributions: The paper proposes to add an entropy regularization term in the min-max objective function to generate adversarial samples for training models that are robust to large domain shift. The regularization term is motivated from information bottleneck and is simplified to the entropy of the output of a classification model. Experiments verify the effectiveness of such regularization term.

Strengths: - The proposed regularization term is well motivated from the information bottleneck concept. It is simple but effective in improving the model’s robustness in combating domain shift. - The experiment design is extensive and the results are convincing.

Weaknesses: The major weakness is the novelty. This work is a simple extension to the adversarial data augmentation paper [62] via adding an entropy term as the additional regularization term. The main framework is identical to [62], and the derived bounds are special cases of existing theorems in the literature.

Correctness: The method is sound and is expected to make improvement on the existing adversarial data augmentation method. The empirical methodology is convincing.

Clarity: The paper is generally well written. The proposed method is well motivated and the experimental setting is clear.

Relation to Prior Work: Relation to previous work is not very clear. How the domain generalization and data augmentation are connected and evolved is not very clear.

Reproducibility: Yes

Additional Feedback: After reading authors response, although I am still not convinced of the theoretic contribution, I believe the simple and effective nature of the method, and the experiment stand on their own merits. So I raised my score to accept.


Review 2

Summary and Contributions: The paper proposes a data augmentation procedure to improve the robustness of the model against domain shifts and corruptions. The paper starts with information bottleneck, and after a series of approximations, represents the loss function as a minmax formulation over (backprop-friendly) differentiable functions. Numerical simulations are provided on several datasets against benchmarks. Update after Author feedback: I thank the authors for their reply. I have increased my score from 3 to 4 based on the successful experimental results and other reviewer's comments, but I still believe the theoretical results in the paper are lacking: the theoretical results are not novel (and thus not interesting in their own right) and more importantly, they do not explain the understanding of empirical success of the method. The paper makes sweeping claims about the efficiency of the proposed method that are not true: for example, see lines 37-41: the terms 'efficiently' and 'with proof' are used very loosely there, the problem is still non-convex and needs to be solved. Regarding the author's feedback on Theorem 2: Consider the case when |Y| = N; Then Theorem 2 gives \sqrt{N \log N} bound, whereas the bound in Wu and Yang is only O(1). Following on Theorem 2, as a gradient based algorithm is used to approximate the the entropy regularizer, it is neither sufficient nor necessary to approximate the values of entropy but to approximate its gradient: two functions can be close pointwise but with very different gradients and vice versa. It is not clear to me whether theoretical results in this paper explain their results.

Strengths: The paper shows that adding an differentiable regularizer increases the performance of several architectures on some datasets.

Weaknesses: It is not clear what are the main technical contributions of the paper. The paper oversells it theoretical results and the motivation for the proposed regularizer is weak. 1. The paper misrepresents its contributions in terms of cosmetic theorems and lemma. See the points (i) - (iv) below. (i) For example, Theorem 3, which is an obvious corollary of [31, Theorem 1]. In the Appendix Line 70, it is written that ``After extending it to the case when Y is a deterministic function of X, we get the bound in Theorem 3''. I can't see how is the paper 'extending' the result of Kolchinsky et al. Theorem 3 of this paper is nothing but an obvious Corollary of [31]. (ii) In the same spirit, Theorem 2 of this paper applies standard proof technique form the literature instead of citing existing results from the literature. In fact, stronger results are known for the performance of the plug-in estimate of discrete entropy over a finite size alphabet. For example, see the paper Y. Wu and P. Yang, "Minimax Rates of Entropy Estimation on Large Alphabets via Best Polynomial Approximation," in T-IT, vol. 62, June 2016, doi: 10.1109/TIT.2016.2548468. (iii) Theorem 1 of the paper is nothing but a simple observation. (iv) Lemma 1 is stated as minor adaptation of [8, Theorem 1]. The supplementary material does not contain any information about this adaptation. If the adaptation is trivial (same as Theorem 3), the statement of the result should cite the prior work. 2. The motivation for the proposed regularization is somewhat lacking. The main contribution seems to be that the mutual information term is lower bounded by the entropy of the prediction, and thus proposes minimizing Equation (7) instead. The adversarial examples show that the neural networks can be confidently wrong, which is an issue. However, the proposed method wouldn't find those examples while training because the regularizer favors points with high uncertainty. I am not convinced why the proposed method should be more robust than the usual adversarial training. 3. The paper makes sweeping claims about the efficiency and proof everywhere that are ungrounded, for example, Lines 37-41.

Correctness: Although not a focus of the paper, Lines 190-191 are incorrect; I am not even sure what they are meant to convey.

Clarity: 1. In line 82, authors say they will approximate I(X;Z) by \ell_2 penalty, but then never discuss it again. What does that even mean? 2. The total variation distance or L1 distance was not defined anywhere (for Theorem 3).

Relation to Prior Work: The paper highly misrepresents its contributions in comparison to prior work (see the comments above).

Reproducibility: Yes

Additional Feedback:


Review 3

Summary and Contributions: In this paper, they defined a new regularized adversarial data augmentation inspired from information bottleneck principles by introducing a computationally efficient maximum-entropy regularizer. They theoretically analyzed their framework under deterministic and non-deterministic conditions. Their experiments results are promising.

Strengths: - To me, the most important positive factor of this work is that it can be implemented easily. In general, the types methods that are efficient in implementation, even though though they do not degrade the performance in "usually" boost the results are valuable. In addition, this method is computationally efficient. - they theoretically analyzed their method under deterministic and non-deterministic conditions, and provide a theoretic bound for their method. - In general, they paper is well-written and easy to follow. - Their experiment's results are promising - I like the results explanation for PACS

Weaknesses: - in the abstract, the last few sentences are confusing, for example "so that the generated “hard” adversarial perturbations can improve the model robustness during training" is not clear without the knowledge of "hard" beforehand. Rephrasing will be improve clarity,. - in the abstract and conclusion they mentioned that this regularization produce "hard" adversarial perturbations. I would recommend adding a paragraph within the method where they actually mean by "hard" adversarial perturbation. Especially, can it be measured quantitatively? - They performed the experiments on there data, they could experiment with DomainNet or Office Dataset? Which are popular for domain adaptation research. - I noticed that for digit recognition dataset, they only compare their method against [60] and [62], was is time-consuming to implement the methods in table 2 for digits dataset? I would be interesting to compare against other methods for this data (If other researchers released their code, with some tweaks, it's possible to try them on other datasets). - I would recommend defining a name for their algorithm instead of mentioning 'ours' in the Tables and reports.

Correctness: I did not see anything suspicious. their claim sounds correct to me. (I did not go through their theorems proof)

Clarity: yes, very well-written

Relation to Prior Work: yes

Reproducibility: Yes

Additional Feedback: I read authors response, they addressed my concerns. I liked the experiments and their results, even though the theory part might lack of novelty according to other reviewers comments. I decreased my score a little bit, but still it's an accept to me w.r.t their experiments.


Review 4

Summary and Contributions: Traditional adversarial data augmentation is difficult to define heuristics for generating effective fictitious target distributions. Target to address this issue, the authors designed a new regularization technique (called as maximum-entropy) for adversarial data augmentation from the information theory perspective. The proposed methods enlarge predictive uncertainty of the current model. Extensive experiments have shown the effectiveness of the proposed methods

Strengths: [1] This paper is written well, and clearly explain the motivation and main idea. [2] The definition of problem and the explanation of solution are clear. [3] The ablation study and experiments are relatively sufficient, and clearly show the effectiveness of the proposed method.

Weaknesses: I have try my best to read this paper, but it is actually out of my research area. I am sorry that I cannot give more detailed comments or suggestions.

Correctness: Correct

Clarity: Yes

Relation to Prior Work: Yes

Reproducibility: Yes

Additional Feedback:


Review 5

Summary and Contributions: This paper proposes a regularization scheme that can be used to perform adversarial data augmentation. The method aims to provide robustness against unseen semantic domain shifts in the data, as opposed to the more common case of small perturbations or imperceptible changes. The authors validate their results on a number of datasets, showing strong performance throughout.

Strengths: + This paper is quite well-motivated. The topic of generalization to unseen domains is a crucial application of deep learning, and one that is in need of new algorithms that will enable deep learning to be deployed in safety-critical applications. + The proposed regularizer is well-motivated by the information bottleneck principle (however, see related discussion in "Weaknesses") + The results/experiments are the major strength of this work. The authors show that their method improves significantly over many different baselines on a number of different datasets. The attention to the breadth and depth of the experimental section is admirable, and should be a model for future works in this space. Some comments on the experiments below: + Considering that [62] and this paper share the exact same formation except for the regularization term proposed in this paper, the MNIST results are impressive. It seems that simply adding this regularizer improves performance significantly across domains, especially on SVHN. + The fact that this method is comparable to domain adaptation techniques, which get access to unlabeled data from the target distribution, is quite impressive.

Weaknesses: - Referring to "Theorem 1" as a theorem may be too strong a statement. The result follows in a straightforward way from the data-processing inequality, and the proof is essentially one line. Similarly, "Theorem 3" is essentially a restatement of another result, as shown in the supplemental. - And more generally, a large part of the theory in this paper is essentially restated from [61], excluding of course the regularizer on the mutual information (and eventually the entropy fo the predicted distribution). To this end, calling this regularizer "novel" is in some sense misleading. In essence, the result of the formulation, which is derived from the well-known IBP, is that one should use the formulation of [61] and add a penalty on the entropy of the predicted distribution. Minimizing the entropy of a prediction is an extremely common technique in deep learning. Although it is being applied to a relatively new setting (e.g. the setting of [61] where we want to generalize to unseen domains without any data from those domains), the theoretical contribution and novelty of this work may be overstated in the sense that the idea here is to use a common technique on top of the work of [61]. - Why was AdvAug not used in the PACs experiment? It would be important to know whether the idea of [61] does well here too, so we can see what impact the regularization term has.

Correctness: The results seem correct.

Clarity: The paper is well-written.

Relation to Prior Work: While the authors do in several places acknowledge that their results hinge on those of [61], I am slightly concerned about the novelty of this approach. Given that the only notable innovation here was to use the IBP to justify regularizing the entropy of the predicted distribution, I wonder whether this work is novel enough when compared to [61]. Certainly the new experiments are noteworthy and show good performance, but the remainder of the paper borrows heavily from [61] and well-known principles (e.g. entropy regularization). So aside from the experiments, the overall contribution of this paper is somewhat in question in my opinion.

Reproducibility: Yes

Additional Feedback: My final thoughts on this paper are twofold. First, the theory is on the weaker side, given that it is relatively straightforward to leverage the IBP to derive the entropy regularization term and then attach it onto the optimization problem of [61]. From my perspective, this work lacks novelty on the theoretical side and the contribution is not as significant as one would expect having read the listed contributions. Second, the experiments are quite strong. Empirically, it seems that this method works very well in a variety of scenarios. At the end of the day, this is perhaps the most crucial point, in that in practice one often selects the algorithm that performs best. So given that the results are so strong, I would vote to marginally accept this paper, as the usefulness of this regularization term toward providing robustness against unseen shifts seems significant and moreover because this paper addresses an important problem that is in need of new algorithms.

[Author Response · NeurIPS 2020]

We sincerely thank the reviewers for their feedback. We appreciate they find our work is **"well-motivated"** (R1, R4)
and **"valuable"** (R3); the approach is **"effective"** (R1, R4) and **"efficient"** (R3) in implementation and computation;
the empirical methodology is **"convincing"** (R1); the experiment design is **"extensive"** (R1) and **"sufficient"** (R4);
the results are **"convincing"** (R1) and **"promising"** (R3, R4). We also appreciate they find the paper is **"well-written"**
and **"clear"** (R1, R3, R4). We address the main concerns raised by the reviewers in the rebuttal, and will incorporate all
suggestions for changes in the final version. We sincerely hope this will help the reviewers to finalize their judgments.

**Q1. Responses to R1 on differences and relation to prior work of our method.**

*(1) Differences with [62].* Our work differs from [62] in two aspects. (i) We are the first work to investigate adversarial
data argumentation from an information theory perspective, and address the problem of generating "hard" adversarial
perturbations from IB principle which has not been studied yet. (ii) We theoretically show that IB principle can be
bounded by a maximum-entropy regularization term in the maximization phase of data argumentation, which results in
a notable improvement over [62]: $\geq 5\%$ in accuracy on Digits and CIFAR datasets. *(2) "The derived bounds are special*
*cases of existing theorems".* Our bounds are general in the scenario of DNNs, since they hold under deterministic
and non-deterministic conditions which cover almost all types of DNNs. Thus they can be widely applied in the
training of DNNs and are valuable for further research in this field. *(3) "How the domain generalization and data*
*augmentation are connected and evolved"?* Our work studies a more challenging setting where networks are learned
using **one single** source domain compared with conventional domain generalization that uses multiple training source
domains. Adversarial data augmentation are employed to synthesize virtual target domains during training so that the
generalization and robustness of the learned networks can be improved. We will clarify this part in the final version.

**Q2. Clarification and correction on the necessity and clarity of the theorems. (R2)**

We thank R2 for the detailed comments on our theoretical results. We address the necessity of Theorems 1 and 2 and
will improve the clarity of Theorem 3 and Lemma 1 in the final version. *(1) Necessity of Theorem 1.* Theorem 1 is
necessary because it reveals the connection of IB principle and the proposed maximum-entropy regularization during
adversarial data argumentation. It is an important theoretical contribution of our work. *(2) Clarification on Theorem 2.*
Our bound in Theorem 2 is valuable. First, it is simple and sufficient for optimizing our model. Second, for non-optimal
estimators (which is the condition of networks in the maximization phase during data argumentation), the bound in
$[1*]^1$ (Proposition 1) is not stronger than ours. Both [1*] and our bound are dominated by $\mathcal{O}(|\mathcal{Y}|/N)$ in this case. We
will cite and discuss [1*] in the final version. *(3) "Theorem 3 is an obvious Corollary of [31]".* We will reformulate
this theorem as a corollary of [31] in the final version. *(4) Correction on Lemma 1.* We apologize for the miscitation.
Lemma 1 should be a trivial adaptation of [62] (Lemma 1). We will correct and cite it in the final version.

**Q3. "The motivation for the proposed regularization is somewhat lacking". (R2)**

We do not agree. Our motivation has been recognized by R1, R3 and R4. R2 might misunderstand the scope of our work.
We do not study the problem of adversarial defence where "neural networks can be confidently wrong with **adversarial**
**examples**" as indicated by R2. Instead, this work improves the generalization and robustness of networks for unseen
**real images** by adversarial data augmentation. To achieve this, we propose to train networks with "hard" perturbations
against source data by incorporating the IB principle (lines 29-35). Then we theoretically show our regularization term
which favors points with high uncertainty can bound the IB principle in Eqs. (6) and (7). The results on four **real-world**
**datasets** demonstrate the proposed regularization improves the usual adversarial training [62] by a large margin.

**Q4. Correctness and clarity of the paper. (R2)**

*(1) Correctness of lines 190-191.* We will remove this statement in the final version, because the formal and accurate
description of the problem setup has been defined in Theorem 3. *(2) In line 82, authors say they will approximate*
$I(X; Z)$ *by $\ell_2$ penalty, but then never discuss it again.* As stated in lines 94-96, this is a standard technique for training
DNNs in the literature. We have clearly cited [18, 20] and indicated that we followed their implementation in our work.

**Q5. Responses to R3.**

*(1) Add more explanations of "hard" adversarial perturbations.* We will provide more detailed explanations of "hard"
adversarial perturbations in the abstract and a paragraph within the method in the final version. We will also add
figures of training losses used for measuring them quantitatively. *(2) Add DomainNet or Office dataset.* Our setting is
different from domain adaptation since no image from the target domain is used during training. Thus our method is not
comparable to domain adaptation methods on these two datasets. *(3) Add more competing methods for Digits dataset.*
On this dataset, AGG [42] degrades to [60]. PAR [63] is open-source and we will add its result in the final version.

At last, we will rephrase the corresponding parts (R2, R3) and cite related references (R2). The remaining questions
about writing will be carefully addressed. We thank the reviewers for their careful feedback and consideration.

## Footnotes

[1] [1*] Y. Wu and P. Yang. "Minimax Rates of Entropy Estimation on Large Alphabets via Best Polynomial Approximation".


[Meta-Review · NeurIPS 2020]

The paper was extensively discussed among the reviewers. The final outcome was that all the reviewers agreed that the theoretical part of the paper is not significantly novel and the authors have to rewrite that part (please see the updated reviews), however, the approach is novel and experimental part is strong. To evaluate the experimental part further, a new reviewer was added after the rebuttal who has a good understanding on the experimental side of the topic of adversarial data augmentation. The new reviewer confirmed that the usefulness of the entropy-based regularization term toward providing robustness against unseen shifts is significant.